# The Immune Response Generated against HPV Infection in Men and Its Implications in the Diagnosis of Cancer

**DOI:** 10.3390/microorganisms11061609

**Published:** 2023-06-18

**Authors:** Lilia Chihu-Amparan, Adolfo Pedroza-Saavedra, Lourdes Gutierrez-Xicotencatl

**Affiliations:** Center of Research for Infection Diseases, National Institute of Public Health, Cuernavaca 62100, Morelos, Mexico; lchihu@insp.mx

**Keywords:** human papillomavirus (HPV), men, infection, immune response, cancer

## Abstract

Human papillomavirus (HPV) infection is associated with precancerous lesions and cancer of the genital tract both in women and men. The high incidence of cervical cancer worldwide focused the research on this infection mainly in women and to a lesser extent in men. In this review, we summarized epidemiological, immunological, and diagnostic data associated with HPV and cancer in men. We presented an overview of the main characteristics of HPV and infection in men that are associated with different types of cancer but also associated with male infertility. Men are considered important vectors of HPV transmission to women; therefore, identifying the sexual and social behavioral risk factors associated with HPV infection in men is critical to understand the etiology of the disease. It is also essential to describe how the immune response develops in men during HPV infection or when vaccinated, since this knowledge could help to control the viral transmission to women, decreasing the incidence of cervical cancer, but also could reduce other HPV-associated cancers among men who have sex with men (MSM). Finally, we summarized the methods used over time to detect and genotype HPV genomes, as well as some diagnostic tests that use cellular and viral biomarkers that were identified in HPV-related cancers.

## 1. Introduction

According to the World Health Organization, approximately 630 million people are infected with human papillomavirus (HPV) worldwide, with an estimated prevalence of 9–13% [1], with an estimation of 30 million HPV infections diagnosed each year worldwide [2]. In 2018, data from the GLOBOCAN database showed that 13% (2.2 million) of all new cancer cases worldwide were attributable to viral infections (excluding non-melanoma skin cancers), of which 31.5% (694,000 new cases) correspond to HPV. In men, 2.2% of new cancer cases, and 31.9% in women, are associated with an HPV infection. Of the new cases of cancer associated with HPV infection, in men, 49% correspond to oropharyngeal cancer and 26% to penile cancer; in women, 82% correspond to cervical cancer and 4.2% to squamous anal cancer [3].

The HPVs are members of the *Papillomaviridae* family, and there are five genera (alpha, beta, gamma, mu, and nu) with more than 100 genotypes [4,5]. The HPV alpha genus infects cutaneous and mucosal epithelia and is associated with most cases of cancer [6]. These viruses are small (50–60 nm) with non-enveloped icosahedral capsids that contain three oncoproteins, E6, E7, and E5, that play an essential role in cellular transformation. The E6 oncoprotein induces p53 degradation, inhibits cell cycle arrest and apoptosis, and disrupts essential cell functions [7] to ensure DNA replication [8] and the survival of cells with severe DNA damage [9]. The E7 oncoprotein induces ubiquitination and degradation of the retinoblastoma protein (pRb) to maintain proliferative signals [10]. E7 also functions as a promoter for cell growth and replication by inducing the S phase through the activation of cyclin A and cyclin E and the inactivation of the cell cycle regulators, p21^Waf1^ and p27^Kip1^ [7]. HPV16 E5 delays the differentiation of keratinocytes and participates in the evasion of the immune response against HPV. E5 can evade the immune response by reducing the antigen presentation by inhibiting the transport of the MHC-I to the cell surface. This event produces a decreased recognition of HPV-infected cells by specific CD8^+^ T cells. Regarding MHC-II proteins, E5 prevents the activation of these molecules by inhibiting the degradation of the invariant chain, resulting in its intracellular accumulation and unable to load processed peptides for antigen presentation [11]. E5 can promote cancer cell proliferation by interacting with the epidermal growth factor receptor (EGFR). Additionally, E5 inhibits apoptosis by increasing ubiquitination and proteasome degradation of the pro-apoptotic protein Bax [7], and the degradation of Fas receptors preventing the death domain formation [11]. E6 and E7 disrupt interferon and NFκB signaling pathways, allowing the virus to persist and escape detection by the immune system [6]. These events induce the proliferation, immortalization, and malignant transformation of HPV-infected cells [9].

After infection, the HPV genomes are established as episomes. In this stage, the oncoproteins E6 and E7 are produced and regulated by the E2 protein through binding to the long control region (LCR) of HPV. During a productive infection that can lead to tissue lesions, the virus is maintained at a high number of episomes that depend on the cells’ differentiation program. Later on, during the progression from high-grade squamous intraepithelial lesions (HSIL) to cancer, the HPV genome generally is integrated into the cellular genome. This integration, where the circular DNA is disrupted between the E1 and E2 viral genes, leads to the loss of E2 expression and then the upregulated production of E6 and E7 oncoproteins. During the viral genome insertion, the expression of E5 is also lost, which initially functions in maintaining cell proliferation through the EGFR. Altogether, these events ultimately lead to a transformed phenotype [12].

In another context, HPV research is mainly focused on women to study the development of cervical precancerous lesions and cancer, how to prevent these, and the application of ablative therapies and novel immunotherapies that aid in its control [13]. A strategy to prevent this is the vaccination of women with HPV vaccines that favors the reduction in cervical cancer. However, not all countries have full vaccination coverage, and only some of them include boys and men. Most health economic models consider that boys’ and men’s vaccination is not cost-effective when a large group of women needs to be vaccinated. Just recently, it was demonstrated that HPV-associated cancers in men at anatomical sites such as the anus, oral cavity, and oro-pharynx have a similar burden to cervical cancer in women. Considering that vaccinating women over the years will generate herd immunity, it is plausible that only unvaccinated men who have sex with women (MSW) benefit from that immunity, while men that have sex with men (MSM) are not protected; this is an issue that should be considered in vaccination models [14].

Furthermore, the immune response against HPV infection in men was not very well described. For this reason, the knowledge of how HPV infection develops in men and how the immune response is generated against it are questions that must be answered to help to control the infections in MSW (who could act as transmission vectors between women). However, also, the MSM group is a vulnerable fraction of the population that can be infected with HPV, which, together with HIV infection, increases the risk of developing lesions and cancer in different anatomical sites.

This review aims to describe what is known about HPV infection in men and their sexual partners, the associated lesions, the immune response generated against the infection, and the vaccines, as well as laboratory tests to detect HPV infection in men.

## 2. Characteristics of HPV Infection in Men

HPV is the most frequently acquired sexually transmitted infection (STI) worldwide and affects both women and men [15]. Based on their pathogenicity, HPV types from the alpha-papillomavirus genus are classified into low-risk (LR) or high-risk (HR) genotypes. High-risk genotypes such as 16, 18, 31, 33, 35, 39, 45, 51, 52, 56, 58, 59, and 66 are considered carcinogenic to human beings and correspond to the alpha 5, 6, 7, and 9 groups. Low-risk genotypes 6, 11, 40, 42, 44, 54, 61, 70, 72, 81, and 89 cause skin warts, condyloma acuminata, and benign lesions and correspond to the alpha 1, 3, 4, 8, 10, and 11 groups [16,17].

According to some reports, 50–80% of young men and women who are sexually active will acquire a genital HPV infection by having unprotected sexual intercourse with a person infected with at least one type of HPV at some point in their lives [18]. Compared with cervical HPV infection, data on the natural history, epidemiology, and diseases associated with HPV infection of the genital tract in men are limited [19]. However, the prevalence of HPV infection may vary widely based on the study population, the techniques used to collect and process the samples, and the anatomical site(s) or specimen(s) collected for HPV detection. Many studies found that the prevalence of genital HPV DNA in sexually active men varies from 1.3% to 86% in studies with different risk groups and geographic regions evaluating multiple anatomic sites or specimens [20,21]. The problem with the high variability of HPV prevalence in men is that there is no standardized anatomical site; for that reason, Nielson and coworkers tested multiple anatomic sites, and with this strategy, it was possible to observe a higher HPV prevalence in men [22]. However, in many cases, men do not know that they have an HPV infection since it can be asymptomatic and the infection is clear on its own, but in the meantime, the men become a risk factor for the spread of the HPV infection to their sexual partners. 

The diagnosis of HPV infection in men mainly occurs when the lesions are apparent. Additionally, uncircumcised men present HPV infection in many genital sites and this was associated with infertility [18]. It was observed that HPV-infected semen was associated with a reduction in cell viability, motility, displacement, cell count, and normal morphology of cells with increased levels of anti-sperm antibodies (ASA) [23]. In particular, HR-HPV-positive men had lower sperm progressive motility (*p* = 0.007) and higher sperm DNA fragmentation (SDF) values (*p* = 0.003) than those HPV-negative [24]. A study conducted in China to test men´s infertility showed that HPV-positive sperm presented a decrease in motility (*p* < 0.001), vitality (*p* < 0.001), and normal sperm morphology (*p* < 0.05) as compared to the group of HPV-negative cases [25]. HPV does not penetrate the sperm as it does in epithelial cells. In this case, the L1 capsid protein binds to a glycosaminoglycan Syndecan-I located on the surface of the sperm head, an effect that could reduce motility [26]. Another study showed a higher percentage of ASA in HPV-positive males than in HPV-negatives showing alterations in motility and concentration [27]. On the other hand, when men were HPV vaccinated to neutralize seminal infection, there was an increment in sperm motility, reduced ASA percentage, and reduced detection of HPV on sperm and exfoliated cells [28].

For these reasons, it is important to develop specific diagnostic tests that allow early detection of HPV infection in men.

## 3. Stage of HPV Infection in Sexual Partners

In recent years, the interest in understanding the burden of HPV infection increased predominantly among heterosexual men [19]. However, since men are the main reservoirs for transmitting genital HPV infection to women [29], their sexual behavior significantly contributes to subsequent uterine cervical diseases [30].

The transmission rates of HPV infection are not easy to quantify for various reasons, one of which is that many couples are not monogamous. However, Bogaards and colleagues generated a transmission model and found an estimated transmission probability of 0.8 for HPV16 and 0.93 for HPV18 among heterosexual partners [31]. Additionally, Nyitray and colleagues estimated the incidence of HPV transmission in heterosexual partners and found that the transmission incidence rate of genital HPV type-specific was 12.3 (95% CI; 7.1–19.6) per 1000 person months for female-to-male transmission and 7.3 (95% CI; 3.5–13.5) per 1000 person months for male-to-female HPV transmission [32]. A similar pattern was also observed by another group, showing that genital HPV transmission from female to male is more common than from male to female [33,34].

Several research groups analyzed the infection of different HPV genotypes between individuals of a couple to establish if the individuals share the same HPV genotypes and how transmissibility could occur. In these studies, a variable number of couples participate, but concordances only can be obtained from couples where the HPV status and the genotype or genotypes are known for both partners (Table 1). These studies have estimated genotype-specific concordance or discordance of alpha-HPV in heterosexual couples. Some male partners included women with an HR- or LR-HPV infection or cervical lesions. The concordance between sexual couples was separated into three categories: complete when both partners had the same HPV genotypes, partial when they share at least one HPV genotype, and absent when they do not share any HPV or only one of the individuals of the couple had HPV. The concordance data came from the articles, but in some cases, when necessary, we made the calculations of those to better understand the HPV infection behavior in the couples (Table 1). 

In this sense, Nicolau and colleagues showed a complete concordance of the HPV group (HR or LR) in 36.7% of matched couples with an HPV positivity of 70% in men [35]. In contrast, Benevolo and colleagues found that 25% of the couples were HPV-positive in both partners. From their HPV test data, we calculated and obtained a complete concordance of HPV type in 16% of the couples, 8% with a partial concordance, and an absent concordance in 76% of matched couples [38]. Using the Parada and colleagues’ genotypes data, we calculated and identified a higher complete (26.5%) and partial (35.3%) concordance, whereas they reported an absent (38.2%) concordance in matched couples [37].

On the other hand, the prevalence of complete concordance for the same viral type in heterosexual couples was variable among studies ranging from 2.27% to 36.70%. Surprisingly, the sexual partners share only partial HPV genotypes or none at all, as opposed to having the same HPV types, as shown by the low positive concordance in most of these studies (Table 1) [21,34,35,36,37,38,39,40,41,42]. 

HPV is a common infection of the anal canal, and the prevalence varies according to sexual behavior. The highest prevalence (47.2%) was among men who have sex with men (MSM), and the lowest prevalence (12.2%) was among men who have sex with women (MSW) [43], except in those men who had a long-term relationship or had few female sex partners in their lifetime [44]. Still, there are limited data on HPV infection in MSM and MSW partners [20].

## 4. Risk Factors and Lesions Associated with HPV Infection in Men

The principal risk factors associated with HPV infection in men include present and past sexual behavior, cigarette smoking, and lack of circumcision [22]. However, the cross-sectional prevalence of anal HPV infection is mainly associated with a history of receptive anal sex and the number of lifetimes and recent sexual partners [45]. Infections with HR-HPV types were associated with having multiple female sexual partners and also the number of male anal sex partners during life [46].

HPV infections are associated with the development of cutaneous, anogenital, and head and neck lesions in men [47,48]. The International Agency for Research on Cancer (IARC) in 2007 concluded that HPV genotypes of the species alpha 16, 18, 31, 33, 35, 39, 45, 51, 52, 56, 59, 58, and 66 are carcinogenic to humans and classify them within Group I based on the fact that there is sufficient evidence of the carcinogenicity of HPV 16 in the cervix, vulva, vagina, penis, anus, oral cavity, tonsil, and oropharynx [48,49].

In several studies carried out in MSM, MSW, and men who have sex with men and women (MSMW) and showing different types of lesions, genotypes of HR-HPV and LR-HPV were detected (Table 2). The most frequent HPV genotypes found in different samples and lesions were types 6, 11, 16, 18, and 51, with a pooled prevalence of 19.6%, 12.1%, 27.0%, 7.6%, and 5.2%, respectively. In genital warts, the most prevalent types are HPV6 and 11, in a range of 11% to 80% [50,51,52]. 

In the case of normal anal samples, HPV types 6, 11, 16, and 18 were the more frequently found (10% to 14%) [53]. However, in HIV-positive men, the presence of other HR-HPV types, such as 31, 35, 39, 45, 51, 56, 59, and 68, are also highly prevalent (8–39%) [53,54,55,56]. When lesions appear in the anal canal (ASCUS, LSIL, HSIL), the HPV types prevalent are 15, 45, and 51 (3–22%) in HIV-negative men, but in HIV-positive men, the HPV types 18, 33, 35, 39 and 56 also became present (21–37%) [44,57,58]. Conversely, in anal cancer, the prevalent HPV types were 16 and 18 (72% and 7%, respectively) [59] (Table 2). 

On the other hand, samples from normal penis showed a high prevalence of HR-HPV types 16, 51 59, 66 (5–8%), in condyloma lesions the LR types 6 and 11 (30–49%), while in penile intraepithelial neoplasia (PIN), HGSIL and penile cancer, the HPV16 was the most prevalent (57%, 80%, and 69%, respectively) [30,60,61,62]. 

Since HPV types were found in the mucosal epithelia, some authors expanded the search for these types in the oral cavity. They found that in the normal oral cavity of men, HPV16 showed a prevalence of 1% overall [63], but in HIV-positive men, this prevalence went up to 60%, as well as other HR-HPV types such as 18 and 39 that were also present (Table 2) [64,65].

## 5. Natural History of HPV Infection in Men

### 5.1. Genital Warts

One of the most common clinical manifestations of HPV infection is genital warts [66], also known as condyloma acuminata (CA), which are considered one of the most common sexually transmitted diseases caused by HPV infection [67]. Although most genital warts are benign tumors in the anogenital region, they represent a psychological stigma for contracting a venereal disease [68]. The prevalence of genital warts reported in males was 0.16–0.2% in a systematic review [69]. The highest rates of genital warts were found to occur among 20–29 years old males [18]. Approximately 20–30% of genital warts resolve spontaneously within 1 to 2 years; however, recurrence is observed frequently [66,70], and a possible explanation is through autoinoculation [70]. Genital warts are highly infectious; approximately 65% of people with an infected sexual partner will develop genital warts [47]. The incubation period for this type of lesion is around 3 weeks to 8 months (average 2–8 months) [71], with a 90% prevalence of the non-oncogenic HPV types 6 and 11 [18]. In the HPV in Men (HIM) study, 4,123 males were enrolled from Tampa, FL, USA; Cuernavaca, Morelos, Mexico, and Sao Paulo, Brazil, and found that 2.7% of them developed incident genital warts and thirty different HPV genotypes were detected. The most common LR-HPV types detected in genital warts were HPV 6 (43.8%) and HPV 11 (10.7%) and the HR-HPV type 16 and HPV 18 were the more prevalent (9.8% and 3.6%, respectively) [51]. In another study that recruited 53 male patients with genital warts, fifteen different alpha-HPV genotypes were detected: HPV 6 was the most prevalent (79.2%), followed by type 11 (16.9%) and 73 (7.5%) [52]. The study by Al-Awadhi and colleagues, which included women found that 33.3% of men with genital warts had HR-HPVs (types 16, 18, 33, and 38) and 66.7% LR-HPVs (types 1a, 2, 6, 7, 9, 27, 27b, 57b, 57c, 65, and 81) [72]. From the results of these studies, it is clear that warts are associated mainly with HPV infections of the low-risk type. However, there are still limited data about the epidemiology of genital warts in men.

### 5.2. Anal Cancer

Globally, in 2018, anal cancer accounted for about 48,541 new cases and 19,129 deaths. In men, the incidence of anal cancer was 20,196 new cases, and 9,618 deaths, accounting for 0.21% of all new cancer cases [73]. In the last 20–30 years, the incidence of anal cancer increased [66]; it is higher in women than in men, mainly in many high-income countries from Europe, additionally to North America and Australia [3]. However, estimated mortality rates in both men and women were found to be similar, 9.6 and 9.5 per 100,000, respectively [74]. 

Anal squamous cell carcinoma (ASCC) is the most common type of anal cancer. In men, 100% of ASCC is attributed to persistent infections by HR-HPVs [3], with HPV16 being the most commonly detected (70–90%) and, to a lesser extent, HPV18 (<10%) [75]. The presence of HIV infection in men increases the risk to develop anal cancer, an effect that was reported in a systematic review, where it was shown that the incidence of anal cancer in MSM was higher in HIV-positive than in HIV-negative men (*p* = 0.011) with a high prevalence of HPV16 (35%) and HPV18 (19%) in HIV-positive and a low prevalence in HIV-negative for HPV16 (13%) and HPV18 (5%) in all MSM. Also, the prevalence of HPV16 in anal cancer was two times higher in HIV-positive than in HIV-negative MSM [76].

On the other hand, a prospective cohort study of HIV-negative MSM found that anal HPV infection in this group is associated with a high risk of HIV acquisition. Thus, the increasing number of HPV types found in HIV-negative men that seroconverted to HIV-positive could reflect a higher number of anal lesions that facilitate HIV acquisition; nevertheless, there are few studies of the participation of HPV in HIV infection acquisition [77]. In this sense, the prevalence of anal HPV infection among HIV-positive men in different countries varied from 73.7% to 91.6%, with a prevalence of HR-HPV types from 39.7% to 82%, and from these, 5.7% to 61% were associated with multiple HPV infections [78]. 

The prevalence of infection with any HPV type in the anus in MSM is 1.5 times higher in HIV-positive than in HIV-negative men (79% vs. 47%). A similar trend was observed with MSW, although the prevalence was 3.6 times higher in HIV-positive than in HIV-negative individuals (43% vs. 12%) [79]. Interestingly, it was observed that in anal samples of HIV-negative men, more HPV types were detected than in HIV-positive men (Table 2).

Several risk factors are associated with the development of anal cancer; among them are genital or rectal warts, persistent infections with HR-HPVs, HIV-positive, chronic immunosuppression, regular receptive anal intercourse, chronic hemorrhoids, syphilis, multiple sexual partners, gender, age, and smoking [80], also being single or never being married could be a risk factor [30]. Additionally, another risk factor in HIV-positive patients is low CD4 cell counts [81]. On the other hand, an inverse association was found between physical activity and anal cancer (*p* = 0.022), which could be considered a protective effect [80].

To search for biomarkers associated with anal cancer, Walker and colleagues (2009) performed an anal cancer study to evaluate molecules that were reported as predictive biomarkers of favorable prognosis or potential targets in different types of cancers [82,83,84]. The study was carried on with women and men HPV-positive, with anal SIL and/or invasive cancer. The researchers analyzed different growth factor receptors, p16, and Ki-67 expression and showed that 96% of invasive cancers co-expressed EGFR, c-Met, VEGFR1, and p16, but not HER2/neu was observed. They also found a correlation between HIV-positive anal cancer with high expression of c-Met and VEGFR1 (*p* < 0.003). These data suggest that these molecules can be used as potential novel targets to predict the development of anal lesions HPV-associated in the future [82,83,84].

Wessely and colleagues also analyzed women and men with anal cancer for expression of programmed death ligand-1 (PD-L1), whose role is to mediate anti-tumor immune responses. This study demonstrated that 64.3% of HPV-positive tumors and 54.5% of HPV-negative tumors express PD-L1, and those patients that express this molecule had better overall survival than patients with PD-L1 negative tumors (*p* = 0.006). Interestingly, a PD-L1 expression ≥ 10% (TPS category 3 and higher) was significantly more often detected in men (50%) than in women (19.4%). TPS refers to a classification of “tumor proportional score” from 1 to 5. These results indicate that PD-L1 expression can be a prognostic factor for survival, although it is irrespective of the HPV status [85].

### 5.3. Penile Cancer

The global incidence of penile cancer accounts for 34,475 new cases and 15,138 deaths in 2018, which is 0.36% of all new cancers in men [73]. Squamous cell carcinoma (SCC) of the penis is estimated to be about 95% of all penile cancers. The well-differentiated and keratinized SCC has a low association with HPV, whereas undifferentiated SCC warty/basaloid cancers are strongly associated with HPV [86,87]. Approximately 50% of penile carcinoma is associated with HR-HPV infection [19]. Among the risk factors associated with penile cancer are sexually transmitted diseases such as HPV infection, HIV infection, genital warts, poor genital hygiene, lack of neonatal circumcision, history of tobacco consumption, age, phimosis, chronic inflammation, lichen sclerosis (LS), penile injury, and obesity [88,89]. The risk factors associated with penile HPV infection are a high number of sexual partners, having a sexual partner with CIN, a history of smoking, and a history of other STIs [87]. 

Peter and colleagues analyzed the integration of HPV16 in a penile squamous cell carcinoma (IC2 tumor) and in a cell line derived from it (IC2 cell line). They demonstrated that HPV16 DNA integrates into the chromosome band 8q24.21 at the MYC locus and observed MYC amplification (8–12 copies) in both the IC2 tumor and in the cell line, and in the case of the latter, MYC was overexpressed (3.4-fold) [90]. A similar integration event of HPV near the MYC promotor was observed in cervical cancer tumors and cell lines [91]. 

In a systematic review, Olsen and colleagues performed a meta-analysis evaluating HPV transcriptional activity and the correlation with the overexpression of p16. The researchers estimated the pooled prevalence of HPV DNA from different studies (HPV types 6, 11, 16, 18, 31, 33, and 45) and the pooled p16 positivity in penile cancer and penile intraepithelial neoplasia (PIN) worldwide. They observed that in penile cancer, the HPV DNA prevalence varies from 11.6% to 100%, but the pooled HPV DNA prevalence was 50.8%, and among patients with PIN, this was 79.8%. In HPV-positive penile cancers, the most common HPV type was HPV16, and the pooled prevalence between penile cancer (68.3%) and PIN (69.8%) were similar. However, the pooled prevalence of p16 positivity was much higher in HPV-positive (79.6%) than in HPV-negative (18.5%) penile cancer. Meanwhile, the overall prevalence of pooled p16 positivity in penile cancer (41.6%) and PIN (49.5%) were similar [92]. These results suggest that the presence of HPV and overexpression of p16 could be early markers for the detection of these types of lesions. 

### 5.4. Head and Neck Squamous Cell Carcinoma

Head and neck squamous cell carcinoma (HNSCC) is a global burden. In men, HNSCC is the fifth most common cancer worldwide. In 2018, this type of cancer accounted for 666,037 new cases and 340,750 deaths in men. This type of cancer is three-fold higher in men than in females worldwide [73], and approximately 15 to 30% are HPV-associated [93]. In contrast, in oropharyngeal squamous cell carcinomas (OPSCCs), at least 70% are related to an HPV infection [94]. In men, the most prevalent HNSCCs are found in the lip and oral cavity, with 246,420 new cases, and the leading cause of HNSCC with 119,693 deaths, accounting for 2.6% of all new cancers [73]. In developed countries, HPV-positive HNSCC risk factors are related to sexual behavior toward a genital–oral relationship, initiation of sexual activity at an early age, and multiple sexual partners [95]. Also, several studies reported that individuals who engage in oral sexual activities have a 4.3-fold increased risk of developing HPV-related HNSCC (*p* = 0.004) [95]. In addition to HR-HPV infection, other risk factors for developing HNSCC are alcohol and tobacco [96]. The HPV DNA prevalence reported by anatomical site was 45.8% for the oropharynx, 22.1% for the larynx and hypopharynx, and 24.2% for the oral cavity [84]. HPV16 is the most common type and is found in approximately 90% of HPV-positive HNSCCs, and less frequently observed are HPV types 18, 31, and 33, as was reported in different countries [97]. It was reported that between 39% to 71% of HPV-positive HNSCC viral genomes integrate into the host genome [98].

On the other hand, HPV-associated viral gene expression was evaluated in HNSCCs. Parfenov and colleagues found a higher expression of E2, E4, and E5 than of E6/E7 when HPV is not integrated into the cellular genome [99,100]. Once the viral genome integrated, a high expression of E6/E7 and loss of E2, E4, and E5 expression was observed, allowing the carcinogenic process to initiate (Table 3) [12].

Cellular gene alterations were also significantly studied in HPV-positive and negative HNSCC and performed with samples from women and men. The gene alterations reported include gene expression alterations, mutations, deletions, copy number amplification (CNA), and fusions (Table 3). Some percentages shown in Table 3 were obtained by using the cBio cancer genomics portal (http://cbioportal.org accessed on 14 April 2023) with data from HPV-positive HNSCCs of The Cancer Genome Atlas [100,103].

Genes exhibiting increased expression in HPV-positive HNSCC tumors are associated with cellular events such as cell cycle, proliferation, migration, differentiation, apoptosis, DNA replication and repair, innate immune response, and tumor-suppressing (Table 3). In this context, the FGFR3 gene, associated with differentiation, presents mutations and the FGFR3-TACC3 gene fusion. This fusion was present in the same proportion as the mutations of FGFR3 [103], and it was reported previously in other types of cancer but not found in normal tissues [100,107]. Thus, the codified fusion protein could be a potential prognostic/predictive biomarker for selecting and monitoring patients for FGFR-targeted therapies and identifying resistance biomarkers [108]. Other genes with alterations in CNA are the proto-oncogenes PIK3CA, SOX2, and FGFR3. In the case of SOX2, the detection of the protein in patients proved to have a better survival prognosis. Therefore, Chung and colleagues proposed that the SOX2 signature can be used as a prognostic marker of HNSCC [109].

Another important gene implicated in proliferation and differentiation is the EGFR which is altered by mutations, CNA, and deletions [100,103,104]. These types of alterations in oropharyngeal tumors showed high EGFR levels together with HPV16 E5 expression (*p* = 0.03) and also identified that high E5 levels are associated with recurrence-free survival (*p* = 0.02), although high EGFR levels correlate with decreased recurrence-free and overall survival (*p* < 0.001 and 0.006, respectively). Furthermore, a comparison of recurrence-free survival between high levels of E5 and EGFR was inversely proportional [110]. 

Several tumor suppressor genes present mutations, deletions, and CNA. P53 mutations were found in various tumor cells, causing inactivation and loss of tumor suppressor protein functions, affecting targets such as the cell cycle inhibitor p21, DNA-repair proteins POLK and GADD45, pro-apoptotic proteins BAX and the proliferation-related protein P48 [111,112]. On the other hand, RB gene mutations produce a protein that dissociates from the pRB-E2F1 complex affecting cell cycle regulation [112]. 

A minor proportion of HPV-positive tumors from the TCGA data present 14% of recurrent deletions and 8% of truncating mutations of the gene of TNF receptor-associated factor 3 (TRAF3); this inactivation could allow HPV to evade innate and interferon responses. Therefore, some of these genes can be considered candidate therapeutic targets [100]. In addition, immune response genes such as interferon-gamma were found to present mutations, deletions, and CNA in this type of cancer [103].

It was also observed that some genes exhibit down-regulated expression in HPV-positive HNSCC like NAP1L2, which codifies for a protein implicated in nucleosome assembly, and KIRREL, which codifies for the NAPH1 protein that participates in cell-to-cell adhesion (Table 3) [101,102,105,106].

Despite the diverse alterations in key cellular genes observed in HPV-positive HNSCCs, more detailed research is needed to establish the sequence of the events by which these alterations occur after HPV infection and that finally allow the development of this type of cancer.

## 6. Natural HPV Immune Response in Men

The role of immune responses, cellular and humoral, is crucial to determine whether an HPV infection progress to precancerous lesions or cancer. Several factors minimize the exposure of HPV to the immune system since the virus does not infect or multiply in antigen-presenting cells located in the epithelium, does not lyse keratinocytes, and does not cause viremia [18]. Similarly, since there is no animal model close to the human in which to study the immune response against HPV, several studies were conducted in humans, in which blood or biopsy samples from HPV-infected individuals helped to determine the role of the immune system against HPV.

From those studies, it was identified that the cells involved in the innate immune response against HPV are natural killer (NK) cells, which are generally present in virus-associated lesions and preneoplasic lesions. However, it was observed that there is resistance of the cells of cancerous lesions to be attacked by NKs, which at the same time present a decreased and restricted activity of immunostimulatory cytokines [113,114]. 

Other cells involved in the response against HPV are T lymphocytes, especially cytotoxic T lymphocytes (CTLs), which were attributed to the elimination of virus-infected cells [115,116] and the regression of lesions caused by infection [113,117], especially in regressive cutaneous warts, where they are frequently seen with lymphocytes T CD4+ and macrophages [117,118]. In this case, activation of Th1 lymphocytes allows a better response against viral infection [117]. However, in advanced infections, Th1 activation can be neutralized by the increased secretion of the IL-2 receptor or Th2 over-response [116,119]. Although in men, the cellular immune response to HPV infections was less studied than in women, it was demonstrated that Th1 cell-mediated cytokine response (INF-γ and IL-2) was associated with natural HPV clearance [120]. Likewise, the importance of the response of T-lymphocytes in the elimination of HPV infection was demonstrated in HIV-positive men and women who present a high frequency of recurrent infections and viral persistence in the skin, genital warts, as well as cancers of the uterine cervix, anus, or penis [117]. 

HPV infections are more frequent in the MSM population due to the higher prevalence of men with HIV compared to those from the MSW population. In the MSM-HIV positive group, there is an increment in the prevalence of HPV as well as in the number of types detected in the anal canal when the CD4+ T cell counts decrease, which suggests that CD4+ T cells play an important role in the control of HPV infection in this group; hence, broad surveillance is highly recommended [121,122]. When individuals are treated with antiretroviral drugs such as NVP+3TC+AZT (Nevirapine, Lamivudine, and Zidovudine), there is an increase in CD4+ cells and, at the same time, a reduction in HPV infection. However, men taking EFV+3TC+TDF treatment (Efavirenz, Lamivudine, and Zidovudine) did not show a significant increase in CD4+ cells or clearance of HPV infection. This confirms that CD4+ cells are important in eliminating HPV infections [123].

To closely determine the CD4+ T-cell response in the MSM group with anal lesions, peptides derived from HPV16 antigens were used, and they identified that 59.7% and 29.9% of the individuals presented a specific CD4+ T-cell response against E6 and E7, respectively, whereas E6- and E7-specific CD8+ T-cell responses were both around 19%. In this study, the researchers demonstrated that E6-specific CD4+ T-cell response is associated with HSIL regression [124].

A meta-analysis carried out by Litwin and colleagues (2020) to evaluate infiltrating T-Cell markers in cervical carcinogenesis demonstrated that there were lower levels of CD3+, CD4+, and CD8+ cells in uterine cervical lesions than in cancer as compared to normal epithelium. Also, it was observed that the presence of FoxP3+ and CD25+ regulatory tumor-infiltrated T-cells was higher in persistent and precancerous lesions than in uterine cervical cancer. Regulatory T-cells are considered inhibitory of the immune response, so improved outcomes with lower regulatory T-cell levels. Successful evasion of the immune response is required for HPV infections, where infiltrating (CD8+, CD4+) T-cells are reduced and regulatory (FoxP3, CD25+) tumor-infiltrated T-cells increase, allowing cervical disease progression [125]. However, for men, data for the presence of regulatory T cells are not available, an area that becomes of great interest for anal, oral, and penile cancer in this group.

On the other hand, when looking at the humoral immune response against HPV, this is mainly genotype-specific and is directed to the structural viral capsid protein L1, and those are important for virus neutralization [126]. Antibodies against HPV early proteins also are generated during viral infection in the general population, and these antibodies are suggested to be important as markers of the degree of uterine cervical lesions. In this respect, in women, antibodies against E6 and E7 proteins were associated with CC, antibodies against E4 and E7 are associated with CIN3/CC, and antibodies against E4 are associated with CIN1-2 [127]. 

In the case of the humoral immune response in men, it was observed that higher levels of serum antibodies against virus-like particles (VLPs) from HPV6 and 16 were detected in MSM having anal HPV infection with or without genital coinfection than MSM having only genital HPV infection. Also, this study demonstrated a strong association between seropositivity to HPV6 and the detection of HPV6 DNA in the anal canal among MSW and MSM. However, seropositivity to HPV16 and the presence of HPV16 DNA in the anal canal were associated only with the MSM group [128]. 

Little is known about the antibody response against HPV early proteins in men, but one of those studies showed that in a high-risk men population, a high prevalence of E4 antibodies was present in the urethra and associated with the presence of HPV16 DNA, while the presence of E7 antibodies was high when the HPV16 infection was present in the penis [129]. Another study carried out in Australia with the MSM group showed that HPV16 E6 antibodies and HPV16 DNA detection were associated with the presence of anal HSIL [130]. However, in an HIV-positive MSM longitudinal study, serum antibodies against E6, E7, E1, E2, and L1 from seven HR-HPV were evaluated as predictors of anal HSIL, but no association was observed with any of these markers [131]. In the case of OPC, D’Souza and colleagues (2023) studied the presence of serological biomarkers in a high-risk men population living with HIV and found that anti-E6 antibodies were significant predictors of being at risk to develop oral cancer [132]. The HPV serological antibodies seem to be good predictor biomarkers for HPV-associated diseases in men (anal HSIL and OPC), but more studies need to be carried out to validate the utility of E6 and E7 antibodies as markers of detection and/or progression of these diseases. 

## 7. Vaccines against HPV and the Immune Response Generated in Men

A vaccine derived from recombinant technology is the prophylactic HPV vaccine, which was shown to prevent human cancer development [126]. Three prophylactic HPV vaccines were licensed since 2006 to prevent HPV infection. In 2006, the US Food and Drug Administration (FDA) approved the use of Gardasil, a quadrivalent HPV vaccine (4vHPV) (Merck & Co. Inc, Rahway, NJ, USA) to protect against HPV 6, 11, 16, and 18 infections [133] in females aged 9 to 26 years. Gardasil is a vaccine used to prevent genital warts, in addition to precancerous lesions and cancers from the vagina, vulva, cervix, and anus caused by these HPV types [133]. Subsequently, in 2009, the FDA approved the use of Gardasil in males in the same age group to prevent genital warts caused by HPV types 6 and 11 and anal cancer caused by HPV 6, 11, 16, and 18 [133]. Also in 2009, the FDA approved Cervarix (GlaxoSmithKline, GSK), a bivalent vaccine (2vHPV), for the prevention of cervical pre-cancer and cervical cancer associated with oncogenic HPV types 16 and 18 recommended for girls and young women aged 10–25 years old [133]. However, in 2016, GSK announced the discontinuation of Cervarix due to low market demand.

The third vaccine approved was the nonavalent HPV vaccine (9vHPV) (Merck & Co. Inc) in 2014, which contains the same four types and includes five HR-HPV types 31, 33, 45, 52, and 58. The 9vHPV was originally approved for girls 9–26 years old, and in 2018, the FDA extended the age of approval to be used in women and men from 27 to 45 years old [134]. Gardasil quadrivalent was replaced with Gardasil 9, and it was discontinued in 2017 in the USA (https://www.cdc.gov/vaccines/hcp/vis/what-is-new.html, accessed on 14 April 2023).

The benefit of the HPV vaccine in reducing female cancer was well studied, and the mathematical models for cervical cancer indicate that the effectiveness of the vaccine in men is low and without much benefit in reducing the disease and is considered not cost-effective. However, men develop other HPV-associated cancers, such as anal, oropharyngeal, and from oral cavity, which are not necessarily covered by herd protection, especially the MSM population that remains positive for HPV-associated diseases [135]. 

There were some studies that addressed the effectiveness of the HPV vaccine in different men populations. In this sense, Hillman and colleagues, in a randomized placebo-controlled, double-blind study, evaluated the immunogenicity of the 4vHPV vaccine in men in a three-dose scheme (0, 2, and 6 months) and reported that ≥97.4% of MSW and MSM aged 16 to 26 years seroconverted for all types of HPV present in the vaccine at month 7 (Table 4). However, the antibody titers decreased considerably by month 36 for all HPV types, where 88.9%, 94.0%, 97.9%, and 57.0% of individuals remained seropositive (HPV-6, -11, -16, and -18, respectively) as compared with basal titers, although the titers remain higher than those observed from natural infection [136]. 

To analyze antibody titers in different studies, we used the geometric mean titers (GMT) values reported in milli-Merck units per milliliter (mMU/mL), and compared HPV serostatus at month 7 of men and women who received three doses of the 4vHPV vaccine. In Table 4, it is shown that for men, the antibody titers for the four HPV types are similar in all age groups (from 16 to 45 years old), while for women who are 16 to 26 years of age, the HPV antibody titers are higher than in women 27 to 45 years of age. In the men in the 16-to-26 age group, the HPV antibody titers ranged from 402 to 484 mMU/mL for HPV6 and 18, and from 2404 to 2615 mMU/mL for HPV16 [136,137,138]. However, antibody titers against all four HPV types in this young men group were lower than those reported for women (HPV6~550 mMU/mL, HPV18~740 mMU/mL, and HPV16~3800 mMU/mL) (Table 4) [137,138,139]. 

Another cohort study in men aged 27 to 45 years old investigated whether the immunogenicity produced by the 4vHPV vaccine (three-dose regimen 0, 2, and 6 months) was similar to that generated in young men (16–26 years). The researchers found that men at month 7 were 100% seropositive for all the HPV types contained in the vaccine. The GMT values (0 and 7 months) for each HPV type in the vaccine were: HPV6 (30–420 mMU/mL), HPV11 (40–517 mMU/mL), HPV16 (40–2229 mMU/mL), and HPV18 (16–300 mMU/mL), where the higher titers were for HPV16 [139]. Thus, the immunogenicity among MSM and young men is similar. 

**Table 4 microorganisms-11-01609-t004:** Comparison of geometric means titers at month 7 (after three-dose treatment) among men and women 4vHPV vaccinated.

Gender	Age (Years)	GMT (mMU/mL) (7 Months)				Reference
		HPV 6	HPV 11	HPV 16	HPV 18	
Men	16–23	484	669	2615	473	[136]
Women	16–23	549	636	3870	741	[138]
Men	16–26	448	624	2404	402	[137]
Women	16–26	536	754	2298	458	
Men	27–45	365	490	2178	296	[137]
Women	27–45	412	538	2212	384	
Men	27–45	419	517	2229	300	[139]

Three-dose 4vHPV vaccine was at 0, 2, and 6 months.

Giuliano and colleagues, working with the same study group of young men aged 16–26 years, observed an efficacy of the vaccine of 65.5% against lesions related to HPV 6, 11, 16, or 18 types in the intention-to-treat population. And in the per-protocol population, an efficacy of 90.4% was observed against external genital lesions development related to HPV 6, 11, 16, or 18 types. The condyloma acuminata and all PIN lesions presented an incidence reduction of 89.4% and 100%, respectively. Although the point efficacy estimates for the boys and men in this study were numerically lower than those for girls and women in previous studies, the confidence intervals overlapped, suggesting that vaccine efficacy may be similar for the two sexes [140].

In another study, Castellsagué and colleagues analyzed the immunogenicity of the 9vHPV vaccine in the group aged 16 to 26 years old with a three-dose regimen in women, MSW, and MSM. At month 7, the authors found a seroconversion of >99% for each of the HPV types included in the vaccine in all participants (women, MSW, or MSM). The type-specific antibody response for HPV types of the 9vHPV vaccine was lower in MSM than in women or MSW, with a GMT ratio ranging from 0.70 to 0.89 (GMT MSM/GMT women) and 0.59 to 0.75 (GMT MSM/GMT MSW), respectively. In comparison with the type-specific antibody response in women and MSW with a GMT ratio ranging from 0.78 to 0.91 (GMT women/GMT MSW). This study did not present vaccine efficacy against the development of lesions in men [141].

Recently, Drolet and colleagues realized a systematic review and meta-analysis of the population-level impact of vaccination on girls, women, boys, and men against HPV infections and anogenital wart diagnoses. They found, in boys aged 15–19 years and men aged 20–24 years, a 48% and 32% significant reduction in anogenital warts diagnoses, respectively. However, according to the multiple cohort vaccination and high routine vaccination coverage after 5–8 years of HPV vaccination, the percentage of diagnoses varies in boys younger than 20 years from 88% to 1% in countries with single-cohort vaccination and low routine vaccination coverage. Therefore, since many were infected with the vaccine HPV types, the effectiveness of the two-dose vaccine regimen in individuals older than 18 years should be considered, but until now, a three-dose schedule is recommended [142].

The few studies on the immunogenicity of the HPV vaccine in men indicate that vaccination against HPV in men will help prevent the development of genital warts and various types of cancer, such as anal, penile, and head and neck. In addition, the vaccination of boys and men against HPV will also protect their sexual partners from developing lesions associated with HPV infection.

## 8. Detection Methods

There is no approved diagnostic test to detect HPV infection in men. The US Centers for Diseases Control and Prevention (CDC) does not recommend screening tests for HPV infection in men in the United States (CDC Fact Sheet. https://www.cdc.gov/std/hpv/stdfact-hpv-and-men.htm, accessed on 29 April 2023). However, it is important to start looking for some specific tests, as more studies showed that men also develop HPV-associated malignancies, and herd immunity is not enough to protect them. These studies used some of the different tests that were approved to detect the presence of HPV in women, but those tests have to be standardized according to the type and the body site from which the samples are taken.

### Molecular-Based HPV Detection Tests

The methods based on the molecular detection of the HPV DNA genomes, which routinely were used in samples from women, were also used to detect HPV infection in men. Generally, all these methods involve the realization of a PCR with consensus primers derived from HPV L1 or E6/E7 genes of various HPV genomes to amplify a fragment and further genotyping through diverse techniques by using specific probes. A summary of some of these tests is presented in Table 5.

One of the first tests to be used in men samples was the PCR using the MY09/MY11 consensus degenerated primer pairs located within the HPV-L1 highly conserved gene. However, due to the lack of information regardless of the sensitivity and specificity of the MY09/11 consensus PCR primers to detect oncogenic HPV types, Depuydt and colleagues used samples from women from the Laboratory for Clinical Pathology to compare the PCR consensus primers with type-specific PCR primers (directed to E6 and E7 genes) for oncogenic HPV types. They found that MY09/11 consensus and type-specific PCR primers present sensitivity of 87.9% and 98.3%, respectively, and a specificity of 38.7% vs. 76.1% to detect biopsy-proven CIN2+ [143]. Based on these results, it was considered that the type-specific PCR primers could be extended to studies in men to detect HPV infection and its relationship with lesions.

**Table 5 microorganisms-11-01609-t005:** Prevalence of HPV in men in different body samples and through different tests.

Type of Sample	Methodology	Prevalence (%)				Reference
		LR-HPV	HR-HPV	HPV16	HPV18	
Genital warts	PCR-reverse dot blot hybridization (GP5+/GP6+)	91.0	41.9	12.9	7.2	[144]
Anogenital warts	qPCR (MY11/GP6, HPV2/B5 primers)	66.7	33.3	N.R.	N.R.	[72]
Penile exfoliated cells	Reverse line blot (Biotinylated MY09/MY11 primers)	29.0	71.0	9.0	N.R.	[129]
Exfoliate penile cells	PCR and dot blot hybridization (PGMY 09/11 primers)	25.0	48.0	9.6	3.9	[145]
Flat penile lesions (FPL)	PCR (SPF10 primers) DEIA/LiPA25 system	43.0	30.0	6.9	5.2	[146]
Invasive penile cancer	PCR (L1C1/L1C2 primers) and RFLP	N.R.	12.0	12.0	N.R.	[147]
Penile cancer	In situ hybridization (ISH) with Ventana HPV III probes	0.0	11.4	N.R.	N.R.	[148]
Distal urethra exfoliated cells	Reverse line blot (Biotinylated MY09/MY11 primers)	40.0	60.0	0.0	N.R.	[129]
Distal urethra	Multiple HPV genotyping-Luminex system with modified general primers (MGP) derived from GP5+/GP6+ primers	10.8	24.5	8.9	3.2	[149]
Anal canal (MSW)	PCR and reverse line blot hybridization (PGMY 09/11 primers)	5.4	6.8	2.2	0.2	[43]
Anal Canal (MSM)	PCR and dot blot hybridization (MY09/MY11 primers)	26.0	26.0	12	4.7	[77]
Anal canal (MSM)	PCR and reverse line blot hybridization (PGMY 09/11 primers)	20.0	27.3	6.3	4.6	[43]
Anal canal HIV-neg (MSM)	Nested PCR (PGMY and GP5+/GP6+ primers)	10.3 *	50.7	23.0	8.1	[150]
Anal canal HIV-pos (MSM)	Nested PCR (PGMY and GP5+/GP6+ primers)	19.7 *	65.3	33.7	16.3	[150]

N.R. Not reported. * Only referred to HPV11.

In a cross-sectional study involving women and men, tissue samples from genital warts were screened for HPV infection using a qPCR assay containing the MY11/GP6 and HVP2/B5 primers to detect and quantitate HPV in genital warts. By using this test, the authors reported a prevalence in men of 33.3% and 66.7% of oncogenic and non-oncogenic HPV types, respectively (Table 5). In comparison, the prevalence of HPV in women was 40.7% for oncogenic types and 59.3% for non-oncogenic types [72]. Interestingly, a high prevalence of HPV16 (32%) in patients (including women and men) with anogenital warts was observed, while HPV6 associated with these lesions had a prevalence of only 13.5% [72]. More recently, a similar prevalence for HR-HPV in genital warts in men (41.9%) was observed in the study of Yuan and colleagues (2023) by using a PCR with the GP5+/GP6+ primers, but the LR-HPV prevalence was higher (91.0%) than the previous study [144] (Table 5).

In other studies, exfoliated cells, flat lesions, and cancer samples were taken from the penis to detect HPV DNA by using different tests such as reverse line blot, in situ hybridization (ISH), and PCR with different sets of primers. In the different studies, the prevalence reported for exfoliated cells was low for LR-HPV (<30%), this increased when flat lesions were present (43%), but they were not present in penile cancer. However, the prevalence of HR-HPVs in the penis was very high in exfoliated cells samples (40% to 71%), and this prevalence decreased in flat lesions and cancer (30% and 12%, respectively) (Table 5) [129,145,146,147,148].

When samples of the distal urethra in men were taken to detect HPV DNA by reverse line blot hybridization assay or multiple HPV genotyping-Luminex system, many a very different prevalence was observed. The reverse line blot showed a prevalence for LR-HPV of 40% versus 10.8% prevalence detected by the multiple-Luminex, and the prevalence behaved similarly when looking for the HR-HPV types (60% vs. 24.5%, respectively) (Table 5) [129,149]. 

Subsequently, a modified version of PGMY09 and PGMY11 primers was used for PCR in some studies because they showed a higher detection rate of HPV infection than MY09 and MY11 primers. In addition, genotyping was performed by xMAP technology with specific HPV-L1 probes. This PCR test was used in samples of the anal canal in MSM and MSW groups and HIV-positive and -negative men. By using this new PCR, the prevalence in the anal canal for LR-HPV was low for the MSW group (5.4%) and increased in the MSM group (20% to 26%), and it did not seem to be higher in HIV-positive men (19.7%). The prevalence of HR-HPV in the anal canal was low for the MSW group (6.8%) and increased in the MSM group HIV-positive (65.3%) (Table 5) [150,151]. In general, the prevalence of HPV was higher in HIV-positive than in the HIV-negative MSM group, indicating a notable association between HPV infection and HIV-positive status in this group [150].

In all these studies, when they looked for the prevalence of type-specific HPV16 and HPV18 was similar independent of the test or the set of primers used. For instance, in warts, penis, or urethra, the prevalence of HPV16 was between 6.9% and 12.9% (except for one study that did not find this HPV type), and for HPV18, the prevalence was from 3.2% to 7.2%. However, in the canal anal, the prevalence contrasts considerably especially in the MSM group. In this case, the lower prevalence of HPV16 was 2.2% in the MSW group, but it went up to 23% prevalence in the MSM group and to 33.7% in those HIV-positive. A similar trend in the prevalence was observed for HPV18, although those were, in general, lower than for HPV16 (0.2% MSW; 8.1% MSM; 16.3% MSM-HIV-positive) (Table 5) [43,77,129,144,145,146,147,149,150]. 

On the other hand, when looking at another type of HPV-associated cancer, in the case of HNSCC patients, it is essential to determine the HPV status. According to expert consensus opinions, the College of American Pathologists recommended routine HPV testing as a critical parameter for evaluating the clinical outcome of patients [152]. The current gold standard is the E6/E7 mRNA detection, which can be relevant because it allows measuring viral load (VL). The p16 protein (negative suppressor of the cell cycle) and Ki-67 (proliferation marker) were used as surrogate markers for HPV infection [153]. In this case, p16 IHC and p16 ISH were used as strategies for determining HPV status in HNSCC [152]. 

Another novel technic is the droplet digital PCR (ddPCR), which recently was used to determine the VL of E6 and L1 genes in OPC samples from women and men. Stevenson and colleagues, using this technique, found that a medium/high L1 VL was associated with a better clinical outcome, and it was statistically significant (*p* = 0.02). In comparison, medium/high E6 VL showed a slightly better clinical outcome, although this was not statistically significant (*p* = 0.67) [154]. 

The decision to use a specific test for the detection of HPV DNA in men is still not clear, since there are different tests in different formats, which were evaluated in the population; however, there is still no conclusive scientific evidence that allows us to determine which of all the diagnostic systems developed so far can give a more accurate diagnosis and prognosis for the disease. Therefore, a systematic analysis of the different HPV DNA detection tests used in the various population studies worldwide must be carried out and will depend on the reproducibility of the data in the different populations, as well as the sensitivity and specificity of the tests to select the most suitable one, as well as decide on the algorithm to follow in terms of the combination of biomarkers and the treatment to follow with the patients.

## 9. Conclusions 

Research about the relationship between HPV infection on cervical cancer is widely carried out due to the high incidence of this cancer worldwide. Nevertheless, these studies focused on men were carried on to a lesser extent because research becomes more complex due to a reduced incidence of all cancers attributable to HPV in men (being around one-tenth of only the incidence of cervical cancer) [155]. Additionally, the dispersion of studies of HPV-associated cancers at different anatomical sites, such as the penis, anus, urethra, and head and neck, slows down research in this field.

Detection of HPV in men is important as, usually, this group is asymptomatic for the infection. The diagnosis of HPV infection in men only occurs when there are apparent lesions or because of infertility problems; otherwise, men do not have a routinary test such as the Papanicolaou to detect precancerous lesions HPV-associated. Therefore, a selection of an HPV DNA detection test in combination with surrogate biomarkers should be implemented for men for early detection of HPV-associated anogenital lesions.

Men’s vaccination is still under review, as herd immunity carried on from vaccinated women should protect them. However, the MSM group is at high risk of HPV-associated cancers, as the herd immunity from the women group does not function for them. Additionally, vaccination coverage worldwide is still too low to gain this type of immune protection. Then, it is relevant to consider this MSM group in the vaccination models, as this will help to prevent the development of various types of HPV-associated cancers, such as anal, penile, and head and neck. In addition, the vaccination of boys and men against HPV will also protect their sexual partners and avoid the spread of the HPV infection.

## Figures and Tables

**Table 1 microorganisms-11-01609-t001:** HPV genotype concordance in heterosexual couples.

Study Reference	Couples (n)	HPV Genotype Concordance (%)		
		Complete *	Partial **	Absent ***
[34]	25	68.0	NR	12.0
[35]	49	36.7	NR	NR
[36]	238	31.5	68.0	32.0
[37]	34	26.5	35.3	38.3
[38]	25	16.0	8.0	76.0
[39]	114	7.0	0.9	40.3
[40]	29	6.9	24.1	34.5
[21]	23	4.3	52.2	43.5
[41]	88	2.3	21.6	40.9
[42]	34	NR	35.0	NR

NR not reported; * Same viral type; ** At least one viral type; *** Do not share any HPV type.

**Table 2 microorganisms-11-01609-t002:** HPV genotypes detected in different anogenital samples in men.

Sample/Lesion	HPV Genotypes	Prevalence (%)	Reference
Genital Warts	6, 11, 62, **16**, 84, **53**, **52**, 40, **55**, CP6108, **51**, **66**, 42, **59**, **39**, 54, **18**, **58**, **68**, 61, 83, 71, 72, **31**, **45**, **56**, **67**, **82**, **73**, 81	11–44 1–9	[51]
Genital Warts	6, 11 73, **16**, **66**, 62, 84, **53**, **31**, **51**, **55**, 44, **70**, 91	15–80 1–8	[52]
Genital Warts	6 11 73, **16**, **66**, **53**, 62, 84, **31**, **51**, 44, **70**, **55**, 91	76 18 1–7	[50]
Anus: Normal (MSM, MSW, MSMW, HIV positive/negative)	**HIV-negative:** 6, **18**, **16**, 11 **59**, **56**, **82**, **52**, **39**, **58**, **68**, **51**, 61 **66**, **31**, 40, **33**, **45**, **55**, 84, **53**, 44, **35**, 81, 83, 42, 54, **26**, 57 **HIV-positive:** 6, **18**, **56**, **59**, 11, **16** **52**, **82**, **39**, **45**, **51**, **31**, **68** 44, 55, 61, **66**, 40, **58**, **33**, **35**, 83, **53**, 42	10–14 3–9 0.1–2 12–16 8–10 1–7	[53]
Anus: Normal (MSM, HIV-positive)	6, **16**, 11, **18**, **68**, **51**, **52**, 40, **39** **67**, 61, **45**, **58**, **56**, 81, 42, **33**, **53**, 54, 84, **31**, **73**, **82**, 43, **66**, **55**, 44, **35**, 72, **59**, **69**, **70**, 71	3–9 0.2–2	[54]
Anus: Normal (MSM, MSMW, HIV-positive)	**35**, **16**, **70**, 6/11 **58**, **33**, **18**, **56**, **51**, 81, **82**, **68**, **66**, 32/42, **39**, **30**, 71, 61 **85**, **53**, 72, 89/102, **52**, 31, **73**, 83, 84, 62, 90/106, 86/87, **26**/**69**	10–20 4–8 2–3	[55]
Anus: Normal (MSM HIV-positive/negative)	**HIV-negative:** **16**, **53**, 6, **18** 61, **59**, 62, **45**, **58**, **39**, **55**, **51**, **70**, 72, 42, **73** **33**, **56**, **66**, 54, 84, **35**, 11, **68**, 81, **82**, **52**, 83, 40, **69**, **26**, 71, **67**, 64 **HIV-positive:** 6, **16**, **51**, **45**, **35**, **53**, **70**, **39**, **59**, **56**, 84 **68**, **73**, **55**, 11, **82**, 81, **18**, 42, 72, **33**, **52** **58**, **69**, 62, 61, 40, **66**, **67**, 83, 54	18–27 11–17 1–10 25–39 14–22 3–11	[56]
Anus: ASCUS, LSIL, or HSIL (MSM-HIV-positive)	**33**, **16**, **39** **59**, **52**, **51**, **58**, 11, 6	21–30 14–19	[57]
Anus: External Genital Lesions	**16** CP6108, 6, 62, **51**, **59** 53, 84, 61, **66**, 40, **18**, 11	3 1–1.4 0.1–0.9	[44]
Anus: Benign, LSIL or HSIL (MSM, HIV-positive/negative)	**HIV-negative:** 6, **16**, **45**, **51** **56**, 11, **18**, 42, **59**, **67**, **35**, 54, **52** **68**, **39**, **58**, **31**, **33** **HIV-positive:** 6, **16**, 42, 11, **45**, **51** **35**, **67**, **18**, **39**, 54, **56**, **52**, **59**, **68**, **58**, **33**, **31**	12–22 7–11 1–6 21–37 12–19 2–11	[58]
Anus: Anal Cancer (MSW, MSM, MSMW)	**16** **18**	72 7	[59]
Penis, coronal sulcus, glans, shaft, and scrotum: Normal	84, 62, 6, **16**, CP6108, **51**, **59**, 61, **66**, **53** **39**, 81, **52**, 83, 54, **68**, **58**, 70, **56**, 44, **35**, **18** 11, **73**, 40, **31**, 42, 72, **82**, IS39, **67**, **69**, **26**, **33**, 64	5–8 2–4 0.1–1.5	[30]
Penis: Any EGL, Condyloma, Suggestive of Condyloma, PIN, other (MSM, MSW, MSMW)	**EGL:** 6, 11 **51**, **55**, 74, **16**, **31**, 44, **39**, 54 **66**, **26**, **53**, 40, **33**, **18**, 43, **69**/71, **58**, **68**, **82**, **45**, **56**, **73** **Condyloma:** 6, 11 74, **52**, **39**, **51** **16**, **31**, **66**, **26**, 40, 44, 54, **18**, **45**, **56**, **68**, **82**, 9/71 **Suggestive Condyloma:** 6, 11 **51**, **52**, 74 **16**, **31**, **39**, **53**, 43, 44, **18**, **33**, **58**, **66**, 40, 54 **PIN:** **16**, 11 **39**, **51**, 6, **18**, **73**	25–47 2–6 0.3–1 30–49 2–5 0–5–1 18–57 4–7 1–3 29–57 7–14	[60]
Penis: HGSIL	**16** **33**, **58**, **31**, 11, 61	80 1–5	[61]
Penis: Invasive penile cancer	**16** 6, **35**, **45**, **33**, **18**, **52**, **59**, 11, **58**, **73**, **26**, **31**, **39**, **51**, **53**, **56**, **30**, 32, 74, **70**, **66**, **68**, **82**, 27, 40, 42, 43, 76	69 0.3–4	[61]
Penis: Invasive penile cancer	**16**, **18** **33**, 11, **45**, **56**, 42, **31**, **52**, **59**, 6, 43, **58**, **66**, 81, **35**, **51**, **73**	15–78 1–3	[62]
Oral cavity: normal (MSW, MSM, MSMW)	55, **16** 61, 62, **66**, **51**, 6, 71, 72, 84, **70**, **56**, **59**, **35**, **39**, **52**, **58**, 11, **82**, **53**, 64, **69**, 83, CP6108	1 0.1–0.3	[63]
Oral cavity: asymptomatic HIV MSM	**16**, **39** 11, **18**, **52**, **51**, 6, **66** **68**, **70**, 44	60–80 30–54 1–14	[64]
Oral cavity (MSM, HIV-positive/negative)	**HIV-negative:****16**, CP6108, **56**, **45**, **66**, **53**, **33**, **68**, 55, 83, 84, 18, 35, 58, 59, **26**, **70**, **73**, **82**, 6, 11, 54, 62, 72 **HIV-positive:** 55, 72, 84, IS39, **16**, **18**, CP6108 **39**, **51**, **59**, **66**, **68**, **82**, 11, 61, 62, **33**, **45**, **53**, **69**, **70**, 6, 71, **73**, 81	2–4 0.5–1 3 1–2	[65]

MSW: men who have sex with women; MSM: men who have sex with men; MSMW: men who have sex with men and women. EGL: external genital lesion; PIN: penile intraepithelial neoplasia; ASCUS: atypical squamous cells of uncertain significance; LSIL or HSIL: low- or high-grade squamous intraepithelial lesions; HNSCC: head and neck squamous cell carcinoma; HGSIL: high-grade squamous intraepithelial lesions. Bold numbers refer to HR-HPV type: carcinogenic (red), probably carcinogenic (green). Plain numbers refer to the LR-HPV type.

**Table 3 microorganisms-11-01609-t003:** Cellular and viral genes altered in HPV-positive HNSCC.

Gene	Function	Alterations		References
(A) HPV gene expression in HNSCC *				
*E2, E4, E5*	Viral genes	High expression.	Non-integrated vs integrated viral genome.	[99,100]
*E6/E7*	Viral oncogenes	Low expression.		
*E6/E7*	Viral oncogenes	79.6% of total transcripts in HNSCC express HPV viral oncogenes.		[101]
*E2* **	Viral genes	HPV *E2* mRNA expression correlates with relapse-free (*p* < 0.01) and progression-free (*p* < 0.05) survival.		[102]
*E2* mRNA without *E5* expression *	Viral genes	Correlate with a worse prognosis when estimated as progression-free survival but not overall survival of patients.		
(B) Gene alterations in HNSCC * associated with HPV infection				
*PIK3CA*	Oncogene. Proliferation, growth, cell cycle, apoptosis, and cytoskeletal rearrangement.	High expression. Mutations (56%) *** CNA: amplification (27.8%)		[100,103,104]
*SOX2*	Oncogene. Cell fate-determining transcription factor.	*** CNA: amplification 27.8%		[100,103,104]
*E2F1*	Cell cycle progression, DNA-damage response, apoptosis.	High expression. Mutations 19%		[98,100]
*FGFR3*	Oncogene. Growth, proliferation, differentiation, migration, and survival.	Gene fusion/Mutations 11%		[100]
*EGFR*	Cell differentiation and proliferation.	Mutations 6%. *** CNA: homozygous deletion 2.8%		[100,103,104]
*TRAF3*	Innate immune response, apoptotic process.	Mutations 11.1% Inactivation 22% Deletions 3% *** CNA: homozygous deletion 13.9%		[98,100,103,104]
*NOTCH1*	Tumor suppressors.	Mutations 8.3% *** CNA: amplification 2.8%		[100,103,104]
*RB1*	Tumor suppressors.	Mutations 5.6%		[103,104]
*TP53*	Tumor suppressors.	Mutations 3%		[100]
*CDKN2A* (*P16*)	Cell cycle regulator, Tumor suppressors.	Mutations. 5.29 fold high expression		[105]
*RFC4*	Sensor in multiple DNA checkpoint pathways.	3.64-fold higher expression		[105,106]
*CDC7*	Cell cycle regulator.	2.99-fold higher expression		[105,106]
*TOPBP1*	DNA damage response protein.	2.83-fold higher expression		[105,106]
*CDKN2C* (*P18*)	Cell cycle regulator.	2.39-fold higher expression		[105]
*NAP1L2*	Nucleosome assembly.	0.66-fold lower expression		[106]
*KIRREL*	Cell-cell adhesion, excretion, negative regulation of protein phosphorylation, positive regulation of actin filament polymerization.	0.55-fold lower expression		[105,106]

* HNSCC: head and neck squamous cell carcinoma; ** HPV-positive tonsillar and base of tongue squamous cell carcinoma. (SCC and BOTSCC); *** CNA: copy number alteration.

## Data Availability

Not applicable.

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
