# Peer review of "The Immune Response Generated against HPV Infection in Men and Its Implications in the Diagnosis of Cancer"

_microorganisms, 2023, doi:10.3390/microorganisms11061609_

Round 1

Reviewer 1 Report

Authors should be congratulated for the topic. HPV represents a major health concern both for men and for women. While the knowledge of women's pathogenesis and tumorigenesis is deeply understood, men are still shadowed. The manuscript is comprehensive of the main evidence of current literature, it consolidates every aspect of the pathology, of the virus transmission, and of the evolution. Maybe, the fertility field is treated in a scant way and it will be improved (PMID 36576473). However, the manuscript is easily readable and well-written.

Reviewer 2 Report

The abstract was not very clear. I had to read it twice to understand what type of study this was. Rewriting the abstract in a clear, concise manner would help.

It is helpful to explain the methodology of the articles included and excluded in this study. Which search tools were used? Which years? What was the inclusion/ exclusion criteria? A flowchart could illustrate the methodology well.

Otherwise, it was a thorough and well-written review.

Quality of English in this paper was good.
